# Ultrasound Placental Remodeling Patterns and Pathology Characteristics in Patients with History of Mild SARS-CoV-2 Infection during Pregnancy

**DOI:** 10.3390/diagnostics13061200

**Published:** 2023-03-22

**Authors:** Adelina Staicu, Camelia Albu, Georgiana Nemeti, Cosmina Ioana Bondor, Dan Boitor-Borza, Andreia Paraschiva Preda, Andreea Florian, Iulian Gabriel Goidescu, Diana Sachelaru, Nelida Bora, Roxana Constantin, Mihai Surcel, Florin Stamatian, Ioana Cristina Rotar, Gheorghe Cruciat, Daniel Muresan

**Affiliations:** 1Obstetrics and Gynecology I, Mother and Child Department, “Iuliu Hatieganu” University of Medicine and Pharmacy, 400012 Cluj-Napoca, Romania; 2Obstetrics and Gynecology I Clinic, Emergency County Hospital, 400006 Cluj-Napoca, Romania; 3Department of Pathology, “Iuliu Hatieganu” University of Medicine and Pharmacy, 400000 Cluj-Napoca, Romania; 4IMOGEN Centre of Advanced Research Studies, Emergency County Hospital, 400000 Cluj-Napoca, Romania; 5Department of Medical Informatics and Biostatistics, Iuliu Haţieganu University of Medicine and Pharmacy, 400347 Cluj-Napoca, Romania

**Keywords:** prenatal ultrasound, SARS-CoV-2, pregnancy, placental hyperechoic spots, COVID-19, fibrin deposits

## Abstract

**Introduction:** This research aims to describe a progressive pattern of ultrasound placental remodeling in patients with a history of SARS-CoV-2 infection during pregnancy. **Materials and Methods:** This was a longitudinal, cohort study which enrolled 23 pregnant women with a history of former mild SARS-CoV-2 infection during the current pregnancy. Four obstetricians analyzed placental ultrasound images from different gestational ages following COVID infection and identified the presence and degree of remodeling. We assessed the inter-rater agreement and the interclass correlation coefficients. Pathology workup included placental biometry, macroscopic and microscopic examination. **Results:** Serial ultrasound evaluation of the placental morphology revealed a progressive pattern of placental remodeling starting from 30–32 weeks of gestation towards term, occurring approximately 8–10 weeks after the SARS-CoV-2 infection. Placental changes—the “starry sky” appearance and the “white line” along the basal plate—were identified in all cases. Most placentas presented normal subchorionic perivillous fibrin depositions and focal stem villi perivillous fibrin deposits. Focal calcifications were described in only 13% of the cases. **Conclusions:** We identified two ultrasound signs of placental remodeling as potential markers of placental viral shedding following mild SARS-CoV-2. The most likely pathology correspondence for the imaging aspect is perivillous and, respectively, massive subchorionic fibrin deposits identified in most cases.

## 1. Introduction

The COVID-19 outbreak emerged in November 2019 and rapidly became the number one global public health issue [1]. Concerns arose regarding its impact on vulnerable populations, such as pregnant women, maternal–fetal effects, and long-term consequences [2,3]. Despite evidence that vertical transmission of SARS-CoV-2 is unlikely, it is essential to establish potential maternal–fetal short- and long-term consequences of SARS-CoV-2 infection, especially the potential neuro-cognitive and psychologic outcome of the offspring [4,5,6].

Pregnancy-associated severe acute respiratory syndrome coronavirus (SARS-CoV-2) infection seems to prime a generalized inflammatory response associated with hypercoagulability and micro-thrombosis in all maternal organs endorsed by the physiologic procoagulant state of pregnancy [3,7,8]. Since most pregnant women have asymptomatic or mild forms of disease according to the COVID-19 diagnostic criteria proposed by Wu et al., many go through the infection without being confirmed [9]. The succession of events leading to fetal attaint and even stillbirth in the context of COVID-19 infection in pregnancy is probably initiated by viral shedding to the placenta culminating with fetal infection [10,11,12].

Clinicians have been looking to find if placental changes, which may result in a fetal response, are translated by morphologic remodeling patterns that could be detected ultrasonographically [13,14]. This would allow the establishment of follow-up guidance to prevent obstetric complications. Prenatal ultrasound follow-up of placental remodeling has yielded little information concerning potential markers of viral effects [15,16].

The current study proposed the recognition of ultrasound placental remodeling patterns in patients with a history of mild SARS-CoV-2 infection during pregnancy with pathology confirmation as perivillous and, respectively, subchorionic fibrin deposits.

## 2. Materials and Methods

### 2.1. Study Design and Patient Selection

This was a longitudinal, cohort study conducted in the 1st Clinic of Obstetrics and Gynecology, Cluj-Napoca, Cluj-Napoca County Emergency Hospital, Romania between March–December 2021.

The reference group consisted of 67 patients with physiologic pregnancies presenting for prenatal surveillance and delivery in our institution during the above-mentioned time frame, with previous SARS-CoV-2 infection during the current pregnancy, but negative testing at birth. The episode of COVID had been confirmed by positive RT-PCR collected by nasopharyngeal swab during pregnancy and was contracted during the period of the delta wave, but viral typing was not available upon testing. The inclusion criteria were fulfilled by 23 patients. Exclusion criteria were patient refusal to enter the study, smoker status, patient comorbidities, history of/or current preeclampsia, gestational diabetes, intrauterine growth restriction, placental insufficiency, TORCH, viral infections other than SARS-CoV-2 and postdate pregnancy. Information regarding the onset, duration and severity of the SARS-CoV-2 infection, home isolation or hospital admission, and therapy requirement were collected. Maternal pregnancy history, first-, second- and third-trimester ultrasound reports, images, and fetal and neonatal outcome were retrieved from electronic patient records.

### 2.2. Image Analysis

The ultrasound machine used for pregnancy examination was a Voluson E8 Expert with transabdominal GE/RAB2-5-D 3D/4D convex probe 1–4 MHz. Examinations were performed by obstetricians certified in Maternal and Fetal Medicine. Prenatal ultrasound monitoring of pregnancy was achieved according to international guidelines. Placental imaging studies were reviewed by four physicians certified in Maternal and Fetal Medicine. The evaluators viewed static images that they did not acquire personally. All experts, blinded to each other’s results, were asked to identify the presence and degree of the ultrasound remodeling placental pattern. An agreement index was calculated after each expert separately analyzed all the images.

### 2.3. Placental Pathology

Placentas were sent for pathology examination at the Pathology department of IMOGEN Medical Research Institute within ECCHCN, regardless of the delivery route. Before the macroscopic examination, all placentas were immersed in 10% formalin solution. Macroscopic examination and the examination of placental lesions were conducted according to the Amsterdam Placental Workshop Group Consensus Statement [17]. Microscopic examination of a minimum of 5 parenchymal sections and a section of membranes and umbilical cord following Hematoxylin-Eosin stain was performed. Microscopic maternal vascular malperfusion lesions and fetal vascular malperfusion lesions were assessed according to the Amsterdam Placental Workshop Group Consensus Statement [17]. Placental weight and fetoplacental ratio were evaluated [18]. Patterns of perivillous and subchorionic fibrin deposits were mapped and characterized.

### 2.4. Statistical Analysis

Patient data was compiled into a database using Microsoft Excel^®^. SPSS 25.0 was used for the statistical analysis. Data were presented as arithmetic mean ± standard deviation for normally distributed or as median (25th–75th percentile) for those without, with absolute and relative frequencies if the data were qualitative. To appreciate correlations, the Pearson and Spearman coefficients were computed based on the linear or non-linear relationship between the data. The interpretation was made based on Colton rules [19]. To appreciate the inter-rater agreement, we computed the interclass correlation coefficient using the two-way mixed models, type absolute agreement for average measurement with 95% confidence interval (CI) [20]. The alpha error considered was 0.05.

### 2.5. Ethics Statement

The study protocol was approved by the Ethics Committee of the Cluj-Napoca County Emergency Hospital, Romania, number 2079/30.03.2022. Prior to being enrolled in the study, all patients provided their written informed consent elaborated according to the World Medical Association Declaration of Helsinki, including informed consent for neonatal information collection. The reporting of this study conforms to the STROBE statement [21].

## 3. Results

Twenty-three patients fulfilled the inclusion criteria. The mean maternal age at admission was 33.04 (SD ± 5.28 years (25–43 years). Clinical and obstetric patient data were depicted in Table 1. All study patients had a mild form of SARS-CoV-2 infection with mild symptomatology, including fever, cough, anosmia, ageusia, fatigue, myalgia and shortness of breath, and required only symptomatic treatment at home; none had been vaccinated for SARS-CoV-2. All patients had uneventful pregnancies, delivered at term between 38 to 41 weeks of gestation, eutrophic fetuses, with Apgar scores ≥ 8. There were no structural abnormalities in any of the pregnancies evaluated. Obstetric and neonatal outcomes were favorable in all cases, with no significant complications during the neonatal period and up to one year.

### 3.1. Ultrasound Findings Tailored Accord to the Timing of the SARS-CoV-2 Infection

Fetal biometry and Doppler studies were normal, and no structural abnormalities were found in any of the evaluated pregnancies. Sudden onset idiopathic oligohydramnios was signaled at the third trimester morphology scan [30–36] weeks of gestation (WG) in four patients with a history of first-trimester SARS-CoV-2 infection and five patients with second-trimester infection.

Dynamic evaluation of the placental morphology revealed what seems to be a particular pattern of placental remodeling starting from 30 WG towards term following SARS-CoV-2 infection during pregnancy. We noted the occurrence of dispersed hyperechoic foci, without posterior acoustic shadowing, scattered across the placenta, increasing in number and size with consecutive examinations, creating a “starry sky” appearance similar to the sonographic pattern described in acute hepatitis (Figure 1) [22]. Consequently, these foci conflate to form interlobular, chandelier-like, comma-shaped indentations. Later, lesions organize to form a consistent, chalky conglomerate along the entire basal plate, a “white line”, with bolded edges towards the chorionic plate forming white angles (Figure 2). Placental changes mimic the physiologic aging process but occur earlier in gestation; echo-dense foci are more widespread and organize in a short period to form the echoic white line.

Various aspects of placental remodeling is rendered in Figure 3.

Placental ultrasound features were more evident in patients with a history of the second-trimester SARS-CoV-2 infection, observed in 12 of 15 cases, emerging about 8–10 weeks following infection (Table 2). In the subgroup of patients with a history of SARS-CoV-2 infection during the third trimester, both cases with a history of infection onset at 28 WG exhibited these findings.

A negative correlation with statistical significance was achieved between the timespan from the SARS-CoV-2 infection (in weeks) to the first mention of placental maturation and the maternal excess weight gain during pregnancy (r = −0.76, *p* = 0.004).

A negative correlation with some statistical strength was found between the timespan from the SARS-CoV-2 infection (in WG) to the first mention of placental ultrasound changes and small placental weight (r = −0.47, *p* = 0.124) and fetal birth weight (r = −0.316, *p* = 0.318).

Following the evaluation of placental imaging by the four maternal–fetal medicine investigators, the interclass correlation coefficient for average measurements from multiple evaluators with the absolute agreement was 0.68 95% CI [0.41–0.85], *p* < 0.001 for the “starry sky” placental aspect which represents a weak to good agreement. The average score was 1.17 ± 0.57. The inter-observer agreement for the “white line” aspect was 0.90 95% CI [0.81–0.95], *p* < 0.001, representing a good agreement. The average score was 0.54 ± 0.44.

### 3.2. Pathology Findings

The pathology exam found 9 (39.1%) small for gestational age (GA) and 2 (8.7%) abnormally large for GA placentas. It was also found that 60% of placentas from patients with second trimester viral infection were small for GA. Macroscopic examination revealed a peripheric white annular border and circummarginate membrane insertion (Figure 4f).

Maternal vascular malperfusion lesions were noted in all cases, predominantly distal villous hypoplasia and accelerated villous maturation. Fetal vascular malperfusion lesions were less encountered. We rarely found focal calcifications (13%), chorioangiosis (8.7%), and inflammatory lesions (8.7%).

We noted a particular pattern of fibrin deposits around stem villi, represented by various amounts of continuous circumferential perivillous fibrin deposits. When few stem villi were involved the lesion was considered focal, and if most villi were affected, the lesion was considered frequent (Figure 4 and Table 3).

## 4. Discussion

The present study signals potential ultrasound makers of SARS-CoV-2 infection during pregnancy—oligohydramnios, the “starry sky” placental appearance and the basal plate “white line”—which can emerge as risk factors for early placental maturation. This placental pattern is not an instrument to guide obstetric and perinatal management, but it can be an alarm tool mandating caution in pregnancy monitoring and peripartum care. The features appear randomly and seem to respect the same sequence of events.

The earliest signs of advanced placental maturation were noted at 31–32 WG and were exhibited by 80% of patients who experienced SARS-CoV-2 infection during the second trimester of pregnancy. There are approximately 8–10 weeks between the time of infection and the first mention of placental changes similar to the interval from infection to the identification of ultrasound evidence of attaint in the TORCH sequence entities [15].

Similar ultrasound diffuse spot-like echogenic foci involving the entire placenta were previously observed after a first trimester mild SARS-CoV-2 infection, confirmed to be calcifications by the pathology evaluation [13].

Postdate pregnancies are the cardinal exponents of placental calcifications, with placental aging initiating around 34–36 WG and becoming obvious after 40 WG, classically described as the Grannum classification stages [23]. Placental insufficiency may also exhibit progressive calcifications which do not resemble the pattern described in pregnancy 8–10 weeks following SARS-CoV-2 infection. Other factors associated to the occurrence of preterm placental calcifications have been outlined by clinicians, such as smoking, viral infections such as TORCH infections, Parvovirus B19 and Zika [24]. Ultrasound reports of pregnancies complicated by these infections delineate placentomegaly as the most prominent marker of viral shedding, with few mentions of placental calcifications and no description of a specific ultrasound pattern [24]. To avoid all possible bias, smokers, intercurrent viral infections other than COVID, as well as pregnancies complicated by placental insufficiency were excluded from the referential group.

In our study, pathology examination found calcifications in only 13% of the placentas. However, the “starry sky” aspect was described in half of cases, raising the possibility of another histological substrate which could explain imaging findings. The most likely correspondence would be the presence of perivillous fibrin deposits exhibited by 87% of the cases. The lack of a definitive correlation between lesions can be justified by the small number of cases and by the preponderant focal aspect of the perivillous fibrin deposition in our group.

Circummarginate insertion of placental membranes frequently encountered in our study is considered to have an insignificant clinical impact [25], being considered an incidental finding. However, its potential as a marker for SARS-CoV-2 infection during pregnancy should be explored in more extensive studies.

As demonstrated by the good absolute agreement index between our evaluators, these elements are easy to identify during sonographic surveillance of pregnancies. It must be considered that the evaluators viewed static images that they did not acquire personally. In clinical practice, the entire placental volume is evaluated before establishing any diagnosis; therefore, identifying an element should be easier.

Another potential ultrasound distress signal is the sudden third-trimester oligohydramnios encountered in some pregnancies following SARS-CoV-2, observed in two-thirds of patients with first-trimester infection and one-third of those with the second-trimester disease. Oligohydramnios and marked calcifications corresponding to Grannum grade 2–3 in focal zones associated with severe placental insufficiency were previous signaled after a mild, late-second-trimester, A wave SARS-CoV-2 infection [26]. The drop in amniotic fluid quantity, unaccompanied by fetal distress, may be secondary solely to placental damage as illustrated by the pathology report.

Excess maternal weight gain during pregnancy seams to play a role in developing placental injury, especially if the infection occurred during the second trimester. This fact may be explained via the way SARS-CoV-2 enters the host cell using Angiotensin-converting enzyme 2 (ACE2) through S-proteins express on their surface [27].

ACE2 receptors are largely expressed in adipocytes in the white adipose tissue. Its link with diet options, especially high fat diets, further supports the claim that weight gain could lead to high concentrations of ACE2 in pregnant women [28]. This aspect may explain why the ultrasound and histopathological placental injury observed in our study were more critical in the patients that had the infection during the second trimester but had more weight gain during pregnancy than patients with infection during the first trimester but with a normal BMI at the end of pregnancy.

Placental damage was not correlated with the clinical form of SARS-CoV-2 infection. Massive perivillous fibrin deposits were characteristic of placentas from paucisymptomatic COVID-19 pregnant women experiencing sudden intrauterine death [29,30]. Therefore, a sonographic marker signaling a possible major obstetric complication is a useful clinical tool to detect pregnancies requiring more detailed follow-up.

To the best of our knowledge, the present study is the first cohort to report ultrasound placental remodeling patterns, oligohydramnios and pathology characteristics attributed to COVID-19 infection during otherwise normal pregnancy. None of the cases included in our analyses had been vaccinated since patient selection was made prior to January 2022; therefore, no bias factor which could influence viral impact upon placentas was involved. The small number of patients is explained by the strict inclusion criteria to potentially limit known factors responsible for early placental maturation or preterm calcifications and the limited period for selection. Another significant limitation was the selection of patients from a single department and the descriptive status of our study. To confirm our observation, a control group is much needed. However, the ideal witnesses would have to be selected prior to the SARS-CoV-2 pandemic since SARS infections in pregnancy are mostly asymptomatic, and many patients are not diagnosed or not declared. Also, most of the population is now vaccinated for SARS-CoV-2.

## 5. Conclusions

COVID-19 is a widespread contagious disease whose effects on pregnancy are still under observation. Mild SARS-CoV-2 infection during pregnancy may determine advanced placental maturation starting at 31–32 WG and oligohydramnios. We propose a progressive sequence of placenta ultrasound events leading from the ”starry sky” pattern to the continuous basal plate “white line” as a hallmark of placental viral shedding of SARS-CoV-2. The most likely pathology correspondence for the imaging aspect would be the presence of the perivillous and massive subchorionic. Further research on large populations is mandatory to reach safer conclusions.

## Figures and Tables

**Figure 1 diagnostics-13-01200-f001:**
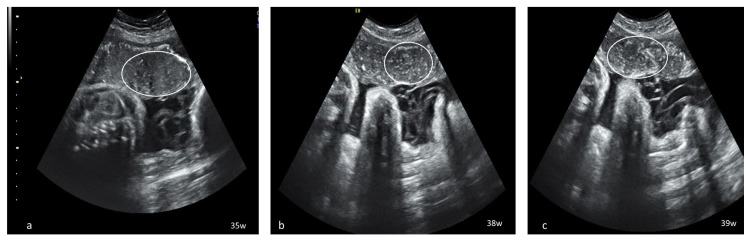
Serial placental ultrasound images at 35 WG, 38 WG and 39 WG, of a nullipara with confirmed SARS-CoV-2 infection at 33 WG (**a**) hyperechoic foci scattered across the placental surface; (**b**,**c**) increasingly denser hyperechoic foci creating a “starry sky” placental pattern (circle).

**Figure 2 diagnostics-13-01200-f002:**
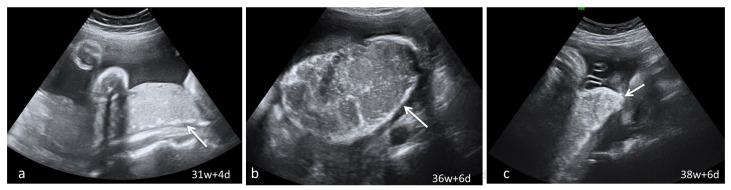
Serial placental ultrasound images at 31 WG, 36 WG and 38 WG, respectively, of a nullipara with confirmed SARS-CoV-2 infection at 24 WG (**a**) the “white line” (arrow) beginning to form at the level of the basal plate; (**b**,**c**) continuous “white line” with bolded angles towards placental margins.

**Figure 3 diagnostics-13-01200-f003:**
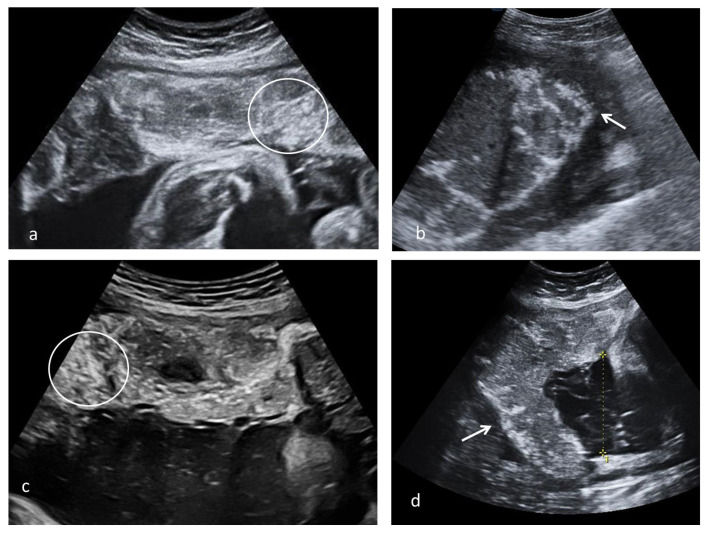
Placental ultrasound placental aspect observed after SARS-CoV-2 infection; (**a**) multiple echogenic foci occupying the entire placental volume and the matte glass appearance in 37 weeks of gestation (WG) second para with SARS-CoV-2 infection at 13 WG, natural birth (circle); (**b**) diffused echogenic foci and continuous white basal line (arrow) in a 39 WG primigravida with SARS-CoV-2 infection at 20 WG, natural birth; (**c**,**d**) multiple echogenic foci occupying the entire placental volume (circle), hypoechoic lacunae and white line (arrow) in a 38 WG primigravida with SARS-CoV-2 infection at 26 WG, caesarian section due to intrapartum fetal distress.

**Figure 4 diagnostics-13-01200-f004:**
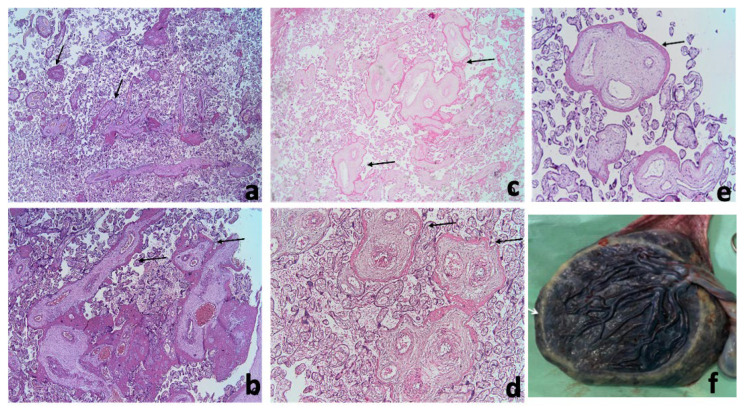
Stem villi perivillous fibrin deposits pattern (arrows–circumferential homogenous pink material around stem villi). (**a**)—focal pattern, most of the stem villi observed are not surrounded by perivillous fibrin deposits (Hematoxylin-Eosin, 2×). (**b**)—frequent pattern, a group of large stem villi and several smaller stem villi surrounded by perivillous fibrin deposits (Hematoxylin-Eosin, 2×). (**c**,**e**)—images from different cases presenting perivillous fibrin depositions around stem villi (**c**)—Hematoxylin-Eosin, 4× (**d**)—Hematoxylin-Eosin, 10×, (**e**) perivillous fibrin depositions around the stem villi Hematoxylin-Eosin, 10×, (**f**) fresh macroscopic, the circummarginate appearance of the placenta.

**Table 1 diagnostics-13-01200-t001:** Clinical and obstetric characteristics of the reference group.

Characteristic	Reference Group (*n* = 23)
**BMI categories, no. (%)**	
Underweight	1 (4)
Normal weight	14 (61)
Overweight	6 (26)
Obese	2 (9)
**Parity, no (%)**	
1	8 (35)
2	8 (35)
>=3	7 (30)
**Timing of SARS-CoV-2 infection, no. (%)**	
1st trimester	6 (26.1)
2nd trimester	15 (65.2)
3rd trimester	2 (8.7)
**Birth weight (g), arithmetic mean ± standard deviation**	
1st trimester SARS-CoV-2 infection	3391.67 ± 272.79
2nd trimester SARS-CoV-2 infection	3660 ± 370.42
3rd trimester SARS-CoV-2 infection	3975 ± n/a

BMI (kg/m^2^)—body mass index; no.—number, n/a—not applicable.

**Table 2 diagnostics-13-01200-t002:** Ultrasound findings according to the timing of viral infection.

Characteristics	Whole Group (*n* = 23)	First Trimester (*n* = 6)	Second Trimester (*n* = 15)	Third Trimester (*n* = 2)
**WG at first mention of placental changes,** Mean ± st. dev.	34.75 ± 2.42	35.8 ± 1.1	34.33 ± 3.01	32 ± n/a
**WG from infection to first mention of ultrasound** placental changes, median (25th–75th percentile)	19.5 (7.5; 30)	30 (30; 32)	9.5 (7; 16)	-
**Amniotic fluid index, no. (%)**				
- oligohydramnios - normal - polyhydramnios	9 (39.1) 12 (52.2) 2 (8.7)	4 (66.7) 2 (33.3) 0 (0)	5 (33.3) 8 (53.3) 2 (13.3)	0 (0) 2 (100) 0 (0)
**Continuous hyperechoic line**				
Data from the first evaluator, no. (%)	14 (60.9)	4 (66.7)	9 (60)	1 (50)
**Reproducibility score,** median (25th–75th percentile)	0.50 (0.13; 1.00)	0.63 (0.00; 1.00)	0.50 (0.13; 1.00)	0.63 (0.25; 1.00)
**Placental hyperechoic areas**				
Data from the first evaluator, no (%)				
-absent	12 (52.2)	3 (50)	8 (53.3)	1 (50)
-focal	2 (8.7)	0 (0)	2 (13.3)	0 (0)
-diffuse	9 (39.1)	3 (50)	5 (33.3)	1 (50)
**Reproducibility score,** median (25th–75th percentile)	1.00 (0.75; 1.63)	1.00 (0.5; 1.75)	1.25 (1; 1.38)	1.25 (0.5; 2)

WG—weeks of gestation; no.—number.

**Table 3 diagnostics-13-01200-t003:** Main pathology findings of the analyzed placentas according to the timing of SARS-CoV-2 infection.

Characteristics	Whole Group (*n* = 23)	First Trimester Infection (*n* = 6)	Second Trimester Infection (*n* = 15)	Third Trimester Infection (*n* = 2)
**Placental weight(g), mean ± st. dev.**	529.04 ± 77.2	491.33 ± 39.81	542.87 ± 84.86	538.5 ± n/a
**Feto-placental ratio, mean ± st. dev.**	6.91 ± 0.78	6.92 ± 0.63	6.84 ± 0.87	7.4 ± n/a
**Maternal vascular malperfusion lesions**, no. (%)				
Distal Villous Hypoplasia	20 (87	5 (83.3)	14 (93.3)	1 (50)
Accelerated villous maturation	14 (63.6)	5 (83.3)	8 (53.3)	1 (50)
**Fetal vascular malperfusion**, no. (%)	9 (39.1)	0 (0)	8 (53.3)	1 (50)
**Overall perivillous fibrin depositions**, no. (%)	18 (78.3)	4 (66.7)	13 (86.7)	1 (50)
Absent	5 (21.7)	2 (33.3)	8 (13.3)	1 (50)
Normal subchorionic	14 (60.9)	3 (50)	10 (66.7)	0 (0)
Massive	4 (17.4)	1 (16.7)	3 (20)	1 (50)
**Intervillous fibrin deposits**, no. (%)	9 (39.1)	4 (66.7)	5 (33.3)	0 (0)
**Stem villi perivillous fibrin deposits**, no. (%)	20 (87)	4 (66.7)	14 (93.3)	2 (100)
None	3 (13)	2 (33.3)	1 (6.7)	0 (0)
Focal	15 (65.2)	3 (50)	11 (73.3)	1 (50)
Frequent	5 (21.7)	1 (16.7)	3 (20)	1 (50)

No.—number.

## Data Availability

All data is available upon request from the corresponding author.

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
