# Peer review of "Ultrasound Placental Remodeling Patterns and Pathology Characteristics in Patients with History of Mild SARS-CoV-2 Infection during Pregnancy"

_diagnostics, 2023, doi:10.3390/diagnostics13061200_

Round 1

Reviewer 1 Report

The manuscript diagnostics-2249865 showed the ultrasound placental remodeling patterns and pathological changes with a history of mild SARS-CoV-2 infection during pregnancy. The theme and aim of the study are very interesting. However, I have several concerns as follows.

1.      A shortcoming of this study is the lack of controls. The authors should compare the placentas of pregnant women not infected with SARS-CoV-2 with those of infected pregnant women.

2.      Figure legend is not sufficiently described. In addition, the presentation of the ultrasound images, especially these are the most important data of this paper, is not clear. The authors should use arrows or other markers to indicate which part is the starry sky or white line.

3.      Figure 3, Markers a) to d) are missing in the figure.

4.      After all, the newborns were not abnormal, so what is the pathophysiological significance of these placental findings?

5.      3.2 line 192. “(Figure3,f)” should be revised.

Author Response

Esteemed Editor and Reviewers,

We received the comments regarding the Manuscript " Ultrasound placental remodeling patterns and pathology characteristics in patients with history of mild SARS-COV2 infection during pregnancy”, and we are grateful for your insightful suggestions and the opportunity to clarify some aspects of our work.

We thank the Reviewers for their enthusiasm for this manuscript, and for recognizing the work as interesting. We have modified the text of the manuscript to address all the critical areas identified by the Reviewers. We hope that our revisions have adequately addressed the points raised by the Reviewers, and that the revised version of the manuscript will be considered acceptable for publication.

In the following section we will detail all the aspects that have been modified in the manuscript:

Response to Reviewer 1:

  1. A shortcoming of this study is the lack of controls. The authors should compare the placentas of pregnant women not infected with SARS-CoV-2 with those of infected pregnant women.

We agree with Reviewer’s comment. An important limitation of this study is the lack of control group and the selection from a single department, as we mentioned in the in the Discussion section. To confirm our observation, a control group is much needed and it represents one of our future ambitions regarding SARS-CoV-2 related research of our department. However, the ideal witnesses (as we mentioned in the article) would have to be selected prior to the SARS-CoV-2 pandemic since SARS infections in pregnancy are mostly asymptomatic, and many patients are not diagnosed or not declared due to fear or stigma, and - most importantly – the vast majority of the population is now vaccinated for SARS-CoV-2, a major bias factor which could influence viral impact upon placentas.

These are the reasons why we limited the present article to a descriptive cohort as a next step following the case reports published in the literature to date.

  1. Figure legend is not sufficiently described. In addition, the presentation of the ultrasound images, especially these are the most important data of this paper, is not clear. The authors should use arrows or other markers to indicate which part is the starry sky or white line.

We used circles and arrows to point out the main ultrasound observations and completed the legend of Figures 1 and 2 as follow:

Figure 1. Serial placental ultrasound images at 35 WG, 38 WG, and 39 WG, of a nullipara with confirmed SARS-CoV-2 infection at 33 WG a) hyperechoic foci scattered across the placental surface; b) and c) increasingly denser hyperechoic foci creating a “starry sky” placental pattern (circle).

Figure 2. Serial placental ultrasound images at 31 WG, 36 WG, and 38 WG respectively, of a nullipara with confirmed SARS-CoV-2 infection at 24 WG a) the “white line” (arrow) beginning to form at the level of the basal plate; b) and c) continuous “white line” with bolded angles towards placental margins.

  1. Figure 3, Markers a) to d) are missing in the figure.

We added the marks to Figure 3.

  1. 4. After all, the newborns were not abnormal, so what is the pathophysiological significance of these placental findings?

„All the newborns from our cohort of patients did have normal obstetric and neonatal outcomes.

However, we must remark that we only included cases with mild forms of SARS-CoV-2 and, even so, maternal vascular malperfusion lesions were noted in all placentas.

Also, massive perivillous fibrin deposits were found to be associated with sudden intrauterine death in mothers who are oligosymptomatic for COVID-19 (Horn, LC., Krücken, I., Hiller, G.G.R. et al. Placental pathology in sudden intrauterine death (SIUD) in SARS-CoV-2-positive oligosymptomatic women. Arch Gynecol Obstet(2022). https://doi.org/10.1007/s00404-022-06614-0).

The available evidence points out that the degree of placental damage is not correlated with the form of SARS-CoV-2 infection and even mild forms of COVID may be complicated with a severe fetal prognosis, even stillbirth. (Schwartz DA, Mulkey SB, Roberts DJ. SARS-CoV-2 placentitis, stillbirth, and maternal COVID-19 vaccination: clinical-pathologic correlations. Am J Obstet Gynecol. 2023 Mar;228(3):261-269. doi: 10.1016/j.ajog.2022.10.001. Epub 2022 Oct 12. PMID: 36243041; PMCID: PMC9554221.)

Therefore, a sonographic marker signaling a possible major obstetric complication is a useful clinical tool to detect pregnancies requiring more detailed follow-up. This information was appended in the Discussion section.

  1. 3.2 line 192. “(Figure3, f)” should be revised.

We apologize and thank you, we corrected to Figure 4, f.

Reviewer 2 Report

This paper aims to describe a progressive pattern of ultrasound placen-19 tal remodeling in patients with a history of SARS-COV2 infection during pregnancy. The authors identified two ultrasound signs of placental remodeling as potential markers of  placental viral shedding following mild SARS-COV2. The most likely pathology correspondence for the imaging aspect is perivillous and, respectively, massive subchorionic fibrin deposits identified in most cases. Statistical analysis is good, well qualified article,  discussed topic is under the current literature.

Author Response

Esteemed Reviewer,

We received the comments regarding the Manuscript " Ultrasound placental remodeling patterns and pathology characteristics in patients with history of mild SARS-COV2 infection during pregnancy”, and we are grateful for your position regarding our work.

We are honored that you appreciate our efforts and consider our article acceptable for publication.

Round 2

Reviewer 1 Report

The authors satisfactorily revised the manuscript. 

The authors analyzed placental findings in COVID-19 unvaccinated patients in the current study. I expect that they will analyze and report on the placentas of pregnant women infected after vaccination in the future.